# Electrophysiological Evaluation of Macular Dystrophies

**DOI:** 10.3390/jcm12041430

**Published:** 2023-02-10

**Authors:** Tsun-Kang Chiang, Minzhong Yu

**Affiliations:** Department of Ophthalmology, University Hospitals, Case Western Reserve University, Cleveland, OH 44106, USA

**Keywords:** electrophysiology, macular dystrophies, Stargardt disease, bestrophinopathies, X-linked retinoschisis, Sorsby fundus dystrophy, Doyne honeycomb retina dystrophy, occult macular dystrophy, North Carolina macular dystrophy, pattern dystrophy, central areolar choroidal dystrophy

## Abstract

Macular dystrophies are a heterogeneous group of genetic disorders that often severely threatens the bilateral central vision of the affected patient. While advances in molecular genetics have been instrumental in the understanding and diagnosis of these disorders, there remains significant phenotypical variation among patients within any particular subset of macular dystrophies. Electrophysiological testing remains a vital tool not only to characterize vision loss for differential diagnosis but also to understand the pathophysiology of these disorders and to monitor the treatment effect, potentially leading to therapeutic advances. This review summarizes the application of electrophysiological testing in macular dystrophies, including Stargardt disease, bestrophinopathies, X-linked retinoschisis, Sorsby fundus dystrophy, Doyne honeycomb retina dystrophy, autosomal dominant drusen, occult macular dystrophy, North Carolina macular dystrophy, pattern dystrophy, and central areolar choroidal dystrophy.

## 1. Introduction

Macular dystrophies (MDs) are a group of inherited retinal disorders that commonly affect bilateral vision. Compared to other inherited retinal dystrophies, MDs are less likely to be associated with nyctalopia, and they often cause varying degrees of bilateral central vision loss. With broad phenotypic variations even among monogenic MDs, the diagnosis, workup, and monitoring of these disorders can be challenging. Recent advances in fundus imaging, optical coherence tomography (OCT), and other imaging modalities, such as fundus autofluorescence (FAF), have improved the diagnosis and management of MDs. OCT and FAF often detect subtle morphological changes in the fundus that is not seen with a fundus exam. While changes in these images may be limited to the central retina, histopathological and electrophysiological changes often reveal diffuse pathological changes [1]. Electrophysiological testing, therefore, also serves as an important tool not only in the diagnosis and monitoring of MDs but also in further enhancing our understanding of the pathobiology of these uncommon disorders.

Different electrophysiological modalities can be used to objectively and non-invasively assess the function of different parts of the visual system. Specifically, for MDs, a full-field electroretinogram (ffERG) can be used to assess the functions of photoreceptors and bipolar cells in rod and cone pathways, while a pattern ERG (PERG) assesses the function of ganglion cell in the central retina. Multifocal electroretinogram (mfERG) allows delineating of multiple local retinal functions in the photoreceptor layer and bipolar-cell layer of the central retina. An electroculogram (EOG) generally identifies abnormalities in the retinal pigment epithelium (RPE) that can be useful in some MDs as well. While other imaging modalities, such as OCT and fundus photo, proivde details on the morphological information of pathologies, electrophysiology testing remains the only way to functionally differentiate MDs from generalized retinal dystrophies.

This review provides an overview of the application of electrophysiology testing in the workup of MDs and discusses the most commonly seen MDs, listed in Table 1. We first describe the epidemiology, inheritance, and clinical presentation and then review the application of electrophysiological testing in the diagnosis and workup for each specific MD.

The diagnosis of inherited macular dystrophies is a complex process that requires full ophthalmologic evaluation, genetic testing, and electrophysiological testing. With the current advances in imaging techniques such as optic coherence tomography (OCT), wide-field fundus imaging, and adaptive optics, electrophysiological testing has not been commonly used in routine ophthalmologic evaluation. However, for inherited retinal disorders, electrophysiological testing remains a critical part of the diagnosis, as it provides an objective evaluation of the function of the visual system, from photoreceptors, bipolar cells, ganglion cells, optic nerve, optic chiasm, to visual cortex. As many inherited retinal dystrophies can present with a high degree of clinical heterogeneity, electrophysiological testing provides an objective means in the assessment of undifferentiated patients who are suspected to have retinal dystrophies.

ffERG measures the generalized electrical response of the retina to diffuse flashes of light from a ganzfeld. The ffERG standard has been issued by the International Society for Clinical Electrophysiology of Vision (ISCEV) [2]. The retinal response is recorded by the recording electrodes on the cornea, bulbar conjunctiva, or lower eyelid skin. The a-wave is the initial negative wave corresponding to the hyperpolarization of the photoreceptors [3]. A-wave reflects the outer retinal function, and its amplitude is measured from the baseline to the a-wave trough. The b-wave is the subsequent positive wave that reflects the depolarization of ON bipolar cells [4], and its amplitude is measured from the a-wave trough to the b-wave peak. The implicit times of a-waves and b-waves are measured from the time point of flash stimulation to the a-wave trough or the b-wave peak, respectively. ffERG is tested in dark adaptation (DA) and light adaptation (LA), respectively. DA protocol is performed after a 20 min of the dark adaptation. With 0.01 cd·s/m^2^ flash stimulus, DA 0.01 ffERG measures the response from the ON bipolar cells of the rod system and shows the b-wave only. With 3 cd·s/m^2^ flash stimulus, DA 3 ffERG presents both a-waves and b-waves, which is a mixed rod–cone response. With a stronger flash stimulus of 10 cd·s/m^2^, DA 10 ffERG allows an estimation of the retinal function even in some special conditions, such as small pupils and opaque media. Oscillatory potentials (OPs) are the wavelets on the ascending limb of b-waves, which are associated with the neuronal activity of the feedback loops from the amacrine cells in the inner retina and are reduced by vascular abnormality [5]. OPs are recorded with DA 3.0 and derived by filtering the components of lower frequencies (e.g., a-wave and b-wave). In the LA condition, LA 3 ffERG mainly measures the responses from cone photoreceptors (a-wave) and ON and OFF bipolar cells (b-wave) after inhibiting the response of the rod photoreceptors with the light adaptation under a background light of 30 cd/m^2^ and eliciting the response by a flash stimulus of 3 cd·s/m^2^ on the same level as the background light. Lastly, the LA 30 Hz flicker, which is elicited by the 30 Hz flicker of 3 cd·s/m^2^ on the background light of 30 cd/m^2^ under the light adaptation, reflects the function of the cone pathway [6].

With digital signal processing technique, multifocal ERG (mfERG) was developed in the 1990s by Dr. Erich Sutter, which assesses the retinal function in the central 40–50 degrees of the retina in the cone pathway under light adaptation [7]. The central retina is stimulated by multiple (e.g., 103 or 61) black/white hexagons, which are alternated according to a pseudorandom binary signal called the m-sequence. The mixed response recorded from the tested eye in a channel is separated with multiple ERG responses by a cross-correlation technique, which correspond to the multiple hexagons/retinal loci [8]. The commonly used parameters of mfERG are the response density (nV/deg^2^) from N1 (the first negative component) to P1 (the first positive component), the N1 implicit time (ms), and the P1 implicit time (ms) of the first-order kernel, which reflect the function of the outer retina of the cone pathway in multiple loci and are more sensitive than ffERG in detecting defects in small areas of retina [6,9].

Pattern ERG (PERG) measures the retinal response to an alternating checkerboard pattern under light adaptation without dilation. Two diagnostic components are obtained from PERG waveform—a positive P50 component around 50 ms and a negative N95 component around 95 ms, in which the amplitudes and implicit times are measured [10]. P50 is contributed by ganglion cells and distal retina, and N95 is mainly associated with the ganglion cell function [6,11]. While PERG is mainly used for testing the RGC function, reports of PERG in macular dystrophies can also be found.

The function of the retinal pigment epithelium (RPE) is associated with the difference in electrical potential between the anterior and the posterior of the eye, known as the standing potential of the eye. The standing potential is associated with the transepithelial potential (TEP) of RPE between the basolateral and the apical membranes. TPE shows a trough during the phase of dark adaptation and a peak during the phase of light adaptation, which may also change when the physiologic status of RPE changes in some diseases. The ratio of the two (light peak to dark trough) is called the LP:DT ratio, previously known as the Arden ratio, and is the primary diagnostic parameter reported by the electroculogram (EOG). The EOG indirectly measures the LP:DT ratio of the TEP according to a protocol published by ISCEV [12], because the eyeball is an electric dipole. For EOG recording, the electrodes are placed at the medial and lateral canthi, respectively. The patient is placed in front of a ganzfeld after a 30- minute of light adaptation in a stable, indoor ambient light. The recording of the 15-minute dark phase starts when the background light is turned off. Afterwards, the 15-minute light phase starts when the background light of 100 photopic cd/m**^2^** is turned on. In both phases, the patient is asked to make horizontal saccades for several cycles per minute according to the on or off of the lights with a fixed viewing angle on the right side and left side from the center in the ganzfeld. The recorded potential between the medial and lateral canthi is changed with the angle of the electric dipole away from the center so that the waveform of saccades can be recorded. A few cycles of saccadic waveforms are recorded every minute and the averaged amplitude at each minute can be obtained. Because the change in this amplitude is proportional to the change in the potential of the electric dipole in the phases of the dark and light adaptation, the LP:DT ratio can be derived. In addition, the fast oscillation of the EOG, which reflects the change in ionic permeability at the apical and basal membranes and the electrical coupling between these membranes by tight junctions in the RPE, can also be measured independently or before the conventional EOG test with different protocols [12].

Using a combination of ffERG, mfERG, PERG, and EOG, electrophysiological tests can be used to characterize and identify potential dysfunction in the visual pathway in macular dystrophies. As MDs typically affect only the central retina, the characteristics of the ffERG cone and rod responses, mfERG, and PERG can help understand the dysfunction in MDs. An abnormal LP:DT ratio of the EOG can be found in MDs with an abnormal RPE. However, the clinical application of electrophysiology has some limitations. As MDs can present at a young age, electrophysiological devices that can help to project the stimulus on the retina and to keep the alignment of the stimulus with the center of retina are needed for PERG and mfERG in young children under anesthesia [13]. Additional normal control values with different types of anesthesia should be established accordingly because general anesthesia may change the electrophysiological parameters [6].

Electrophysiological testing uniquely provide an objective functional test in full ophthalmologic evaluation in the patients suspected of MDs in addition to the other subjective functional assessments, such as visual acuity, contrast sensitivity test, color vision test, and visual field test. When combined with other imaging modalities, such as OCT, electrophysiological testing provides useful information on the potential pathologies, especially in patients without significant abnormality in morphology. In this review, we summarize the specific findings of different MDs in electrophysiology.

## 2. Application of Electrophysiology in Macular Dystrophies

### 2.1. Stargardt Disease

Stargardt disease (STGD1, OMIM: 248200) is the most commonly inherited juvenile macular dystrophy with a prevalence of about 1:8000–1:10,000 [14]. Patients typically present with bilateral central vision loss ranging from 20/70 to 20/200, and the age of onset varies between childhood to early adolescence [14,15]. Late-onset Stargardt disease has also been reported in patients over 45 years of age [16]. An older age of onset is associated with better visual prognosis [15]. This disease is associated with mutations in the ABCA4 gene and has an autosomal recessive inheritance mode [1,15]. Some other genes have also been found to display phenotypes similar to STGD1, such as *ELOVL4* mutations associated with the autosomal dominant Stargardt disease 3 (STGD3, OMIM: 600110) [17,18], and *PROM1* mutations related to the autosomal dominant Stargardt disease 4 (STGD4, OMIM: 603786) [19,20,21]. Some patients with the mutations in *PRPH2* demonstrate significant phenotype overlap with STGD1 and have been thought to have pseudo-Stargardt macular dystrophy with minor differences seen in fundus imaging and OCT imaging [22,23,24].

In the fundus exam, STGD1 is characterized by retinal yellow-white flecks at the level of the RPE, which was formerly known as fundus flavimaculatus. Fluorescein angiography shows a dark choroid pattern due to lipofuscin accumulation at the level of the RPE. While there is no treatment for the disease, patients with Stargardt disease are typically advised to avoid bright light, UV exposure, and vitamin A intake [14].

As there is a large heterogeneity among patients with Stargardt disease, electrophysiological testing is an important tool in classifying the disease severity and monitoring progression. Stargardt disease can be divided into three groups based on electrophysiological findings (Figure 1): in Group 1, the dysfunction is confined to the macula; in Group 2, macular and generalized cone system dysfunction is presented; and in Group 3, macular and both generalized cone and rod system dysfunction are shown [25,26]. This classification is associated with a statistical difference of visual acuity, in which Group 1 has the best visual acuity and Group 3 the worst [26]. Furthermore, ERGs were used to monitor the progression from Group 1 to Group 2 or 3 over time in a cohort study. The primary changes of the ERG over time were reduced amplitudes and delays of the peak implicit time [26]. Similar to STGD1, patients with STGD4 also have reduced amplitudes on the ffERG cone and rod responses (the data were recorded by RETI-port/scan 21, Roland Consult, Brandenburg, Germany) [19,20].

Stargardt disease can also be classified into four types, according to mfERG. Type 1 has a severe reduction in the macular area and reduced/delayed responses in the mid-peripheral area. Type 2 has a reduced but recordable response from the central macula, with more reduced responses in the paramacular area. Type 3 has depressed and delayed responses in the whole tested area. Type 4 has normal responses in the macula and delayed responses in the peripheral area [27,28].

In the EOG, patients with Stargardt disease can present with LP:DT ratio ranging from normal to reduced [29,30]. Specifically, patients with observed flecks are statistically more likely to have a reduced LP:DT ratio, while those without flecks have normal LP:DT ratio [29].

Stargardt disease can also be differentiated into early-onset (≤10 years old at onset) and late-onset (≥45 years old at onset). Early-onset Stargardt disease is characterized by a rapid loss of central vision after onset and early foveal abnormalities. In a study with patients with early-onset Stargardt disease, 58% of patients are in Group 1 with normal photopic and scotopic ffERG, 15% of the patients are in Group 2, and 23% of the patients are in Group 3. However, there was no correlation between the initial ERG classification and the subsequent rate of vision loss in these patients [31]. On the contrary, late-onset Stargardt disease is characterized by a preserved central vision with foveal sparing. While these patients can still have normal or abnormal photopic and/or scotopic ffERG, patients with foveal sparing also show preserved normal responses in the central area of mfERG [32].

Overall, electrophysiological testing provides important information on the evaluation of SD patients and can be used to further distinguish patients into subgroups, as well as to monitor for disease progression in Stargardt disease. With the increase in severity of Stargardt disease, the ffERG changes from normal to decreased cone responses and finally, to decreased cone and rod responses. PERG can also assist in reflecting the function of the cone pathway in another way. mfERG provides another method to classify Stargardt disease, according to the distribution of defective responses across the tested retina. However, electrophysiologic tests are unlikely to differentiate STGD1 from other autosomal dominant phenotypes of Stargardt disease. With the overlapping clinical features and electrophysiological testing between STGD1 and other Stargardt phenotypes (e.g., STGD3, and STGD4), molecular genetic testing is necessary to differentiate these forms.

### 2.2. Bestrophinopathies

Bestrophinopathies are one of the most common macular dystrophies in the world, with an estimated prevalence ranging from 1.5 to 20 people out of 100,000 in various studies [33,34,35], and are associated with mutations in the *BEST1* gene, which has a wide spectrum of phenotypic variations. *BEST1* gene translates to a protein, bestrophin-1, which is a calcium-activated chloride channel in the RPE basolateral membrane. Mutations in *BEST1* gene can lead to the accumulation of lipofuscin that distributes to the RPE and photoreceptor [36]. The four recognized subtypes of bestrophinopathies are all autosomal dominant disorders, except for autosomal recessive bestrophinopathy (ARB) [37].

The most common mutations in *BEST1* is related to Best vitelliform macular dystrophy (BVMD, OMIM: 153700), which is also known as Best disease. It has bilateral macular dystrophy, with the age of onset from 3 to 15 years. The progression of this disease can be described in different stages. Some different classification systems have been proposed based on exam findings [37,38]. In general, the previtelliform stage is characterized by a normal fundus, but may include RPE disruptions in the macular region. This is followed by the vitelliform stage, in which a circular homogenous yolk-like round or oval yellow smooth elevated lesion, with the size of a 0.5-to-2 disc diameter, in macula is observed. The next stage is characterized by the break-up of the previously confluent lesion in the macula at the vitelliruptive stage. Pseudohypopyon can be seen in the next stage with the break-up of the yellow products forming a fluid level in the subretinal space. These changes can lead to the next stage with atrophy and fibrosis. The last stage is characterized by possible choroidal neovascularization and is considered as the final stage associated with neovascularization [38]. ffERG is typically normal in BVMD but mfERG often shows reduced amplitudes in the area with subretinal fluids on optic coherence tomography (OCT), as shown in Figure 2 [33,39]. However, the implicit times in the mfERG are not delayed in BVMD patients [39]. The EOG in symptomatic BVMD is usually abnormal, with a reduced LP:DT ratio of 1.5 or lower, which is most commonly between 1–1.3 (the data were recorded by Monpack One, Metrovision, Perenchies, Francec, and LKC, Gaithersburg, Maryland, respectively) [37,40]. Furthermore, patients with normal fundus exams in the previtelliform stage and some carriers of the *BEST1* mutation do show EOG abnormalities despite normal fundus appearance [34,38].

Adult-onset foveomacular vitelliform dystrophy (AOFMD and AVMD or VMD3 and OMIM 608161) is clinically similar to BVMD. Its difference from BVMD includes that the onset age is older (from 30 to 60 years old) and the central yellow lesion is smaller (from one-third to one disc diameter). The patients typically present without symptoms or can present with metamorphopsia or blurring vision [41]. The lesion gradually enlarges and eventually, can lead to progressive, slow vision loss [42]. It is an autosomal dominant condition resulting from the mutations in *BEST1*, *PRPH2, IMPG1*, or *IMPG2* genes [34,43]. In addition, in contrast with BVMD, EOGs in patients with AOFMD are typically normal, which is suggestive of overall normal RPE functions [41,44]. However, both of the amplitude of the 30 Hz flicker ERG and the central P1 amplitudes in the mfERG are reduced in AOFMD patients, similar to the localized macular dysfunction observed in the mfERG in BVMD patients (the data were recorded by Nicolet Ganzfeld, Nicolet, Madison, WI, USA) [45].

Autosomal dominant vitreoretinochoroidopathy (ADVIRC or VRCP, OMIM 193220) is a rare macular dystrophy that is an autosomal dominant disorder with mutations on *BEST1*, and is typically presented in the first decade of life. The fundus exam is characterized by peripheral chorioretinal degeneration near the ora serrata circumferentially [37,46,47]. In an ffERG, the patients presented with normal to severely reduced photopic and scotopic ERG amplitudes, but they did not show a delay in the implicit time [46,47,48]. In an EOG, a normal to borderline abnormal LP:DT ratio has been reported in ADVIRC patients (the data were recorded by Nicolet Spirit and Ganzfeld, Nicolet, Madison, WI, USA) [47,49].

Autosomal recessive bestrophinopathy (ARB, OMIM 611809) is a *BEST1-*related dystrophy of autosomal recessive inheritance. Clinically, a majority of patients present central vision loss and are often hyperopic [50]. It usually manifests in the first two decades of life [51]. The patients are occasionally associated with a shallow anterior chamber angle that tends to cause glaucoma [51]. Over time, patients with ARB are more likely to develop subretinal fibrosis [37]. In an ffERG, most patients showed a reduction in amplitudes in both the cone and rod responses [50]. In an mfERG, reduced central and preserved paracentral amplitudes were reported in ARB patients [52]. In an EOG, almost all patients with ARB had a severely reduced LP:DT ratio [40,50,51]. The abnormalities in both the EOG and ffERG reflect the dysfunction, not only in the RPE, but also in the overlying retina.

Overall, electrophysiological tests, particularly EOG, provide an important tool to characterize the clinical presentation that us seen in different bestrophinopathies. An LP:DT ratio below 1.5, when compared to the normal value of more than 1.65, defines the abnormal function of the RPE [39,42]. The patients with AVMD usually have a normal EOG, which can be used to differentiate AVMD from all other bestrophinopathies. ARB patients also have reduced scotopic and photopic ffERG in addition to an abnormal EOG LP:DT ratio, which is less common in other autosomal-dominant bestrophinopathies.

### 2.3. X-Linked Retinoschisis

X-linked juvenile retinoschisis (XLRS, OMIM 312700) is the most common inherited disorder affecting macular function in males, with a prevalence between 1:5000 to 1:20,000 [53]. The patients usually present with a bilateral reduced visual acuity at school ages, while some patients can present visual loss in infancy [54,55,56]. The patients often involve serious complications, such as vitreous hemorrhage or retinal detachment [55,57]. In a fundus exam, XLRS is characterized by a split in the inner retinal layers near the fovea in 98–100% of the patients with a central spoke-wheel pattern [53,54]. Half of the patients also have peripheral retinoschisis [53]. XLRS is associated with the *RS1* gene, which translates to retinoschisin, a protein that maintains the structural integrity of the retina.

Electrophysiological testing plays a vital role in helping diagnose the disease along with other diagnostic modalities, such as OCT and fundus autofluorescence. While OCT is currently the mainstay of diagnosis of XLRS, ffERG was the major diagnostic modality prior to OCT [54]. In an ffERG, XLRS patients often have a classical electronegative response to a bright flash in a dark-adapted retina, as shown in Figure 3 [58]. The electronegative response is characterized by reduced b-wave amplitudes to the extent that it is smaller than the relatively preserved a-wave, resulting in a b/a-wave amplitude ratio less than 1.0 (the data were recorded by Nicolet Ganzfeld, Nicolet, Madison, WI, USA) [54,58,59,60,61]. However, the patients presenting a non-electronegative ffERG with a relatively preserved b-wave or reduction in amplitudes in both the a-wave and b-wave have also been reported in multiple studies (the data were recorded by Nicolet Spirit and Ganzfeld, Nicolet, Madison, WI, USA) [56,62,63]. Delayed implicit times of both the a-wave and b-wave have also been reported [64]. Most of the XLRS patients also have reduced on- and off-responses in a photopic on–off ERG [61]. The above ffERG findings in XLRS can be attributed to the abnormalities in the ON and OFF bipolar cells while the photoreceptor function is relatively preserved, which is caused by the schisis between the photoreceptor layer and the bipolar cell layer [65]. In an mfERG, most patients with XLRS, even those with non-electronegative responses in the ffERG, show a significant reduction in the response densities in the retinal area of schisis, while the delayed implicit times are often observed in much larger areas [66].

ffERG and mfERG thus play an essential role in the workup of patients suspected of XLRS in addition to genetic testing. However, electronegative ERG by itself can be associated with other disorders, ranging from other inherited disorders such as congenital stationary night blindness to acquired disorders such as retinal ischemia or paraneoplastic autoimmune retinopathy [67]. A combination of clinical exams, multi-modal imaging including OCT, and fundus imaging as well as electrophysiological testing is therefore necessary for the workup of XLRS.

### 2.4. Sorsby Fundus Dystrophy

Sorsby fundus dystrophy (SFD, OMIM 136900) is a rare autosomal dominant disease that usually affects patients in their fourth to sixth decade [68]. The patients are commonly present with nyctalopia, blurry vision, photopsia, metamorphopsia, changes in color vision, and reduced central vision [69]. The patients can have normal visual acuity in the photopic condition but reduced visual acuity in the scotopic condition, with a delay of the rod-intercept time in dark adaptation testing [70]. In a fundus exam, drusen-like deposits in the macula are the sign of early disease and can progress to sub-retinal hemorrhage, exudates, and eventually geographic atrophy [69,71,72]. It can also lead to acute visual loss from choroidal neovascularization [73]. SFD is associated with mutations in the tissue inhibitor of the metalloproteinases-3 (*TIMP3*) gene that results in the accumulation of extracellular matrix proteins at the level of Bruch’s membrane, which causes amorphous deposits between the RPE basement membrane and the inner collagenous layer of Bruch’s membrane [74,75]. The accumulation of these materials likely contributes to the pathogenesis of SFD. Recent studies have shown that high-dose intakes of vitamin A may improve the results of dark adaptation testing in patients with SFD [70,76].

In the ffERG, patients with SFD may have a reduced rod response and a slightly reduced cone response, but normal amplitudes have also been reported [72,76,77]. It is reported that the EOG LP:DT ratio is either normal or slightly reduced in some cases of SFD [72].

### 2.5. Doyne Honeycomb Retinal Dystrophy (Autosomal Dominant Drusen)

Doyne honeycomb retinal dystrophy (DHRD, OMIM 126600), also known as malattia leventinese, is an autosomal dominant disease. Patients with DHRD present macular drusen early in childhood in the fundus exam, but they remain asymptomatic until their fifth or sixth decade in life [78]. DHRD is associated with a missense mutation, Arg345Trp (R345W), in the *EFEMP1* (epidermal growth factor containing fibrillin-like extracellular matrix protein 1) gene that leads to the accumulation of drusen in the macula between the RPE and Bruch’s membrane from a young age [79,80,81]. As the disease progresses, choroidal neovascularization and large geographic atrophy can result in an acute reduction in visual acuity, metamorphopsia, or paracentral scotoma, which usually occur in the fourth–sixth decades of life [82,83]. Clinically, DHRD is characterized by the drusen found around the optic nerve head, as well as the drusen that forms radial streaks from the center of the macula [80,82]. In an OCT, the patients have diffuse deposits between the RPE and Bruch’s membrane, mostly in the macula with preservation of the neurosensory retinal layers [84].

In the ffERG, symptomatic patients with DHRD can present with normal or reduced b-wave amplitudes in scotopic and photopic responses [85,86,87,88] and a reduced 30 Hz flicker response [82,86]. Oscillatory potentials (OPs) can also be reduced or absent in some patients with DHRD, while most patients have normal OPs. In a PERG, it was reported that there was a mild to a moderate reduction in P50 and N95 components in DHRD patients [87]. In an mfERG, reduced amplitudes can be observed near the macula [85,86]. In an EOG, most DHRD patients have been reported to have responses within a normal range. However, it has been reported in one case that the LP:DT ratio was borderline reduced [82,87].

### 2.6. Occult Macular Dystrophy

Occult macular dystrophy (OMD, OMIM 613587) is characterized by a normal fundus appearance with the progressive loss in vision in both eyes from 6 to 60 years old, with an average onset age of 27.3 years old for visual impairment [89]. After the onset of slightly decreased visual acuity, patients with OMD experience a progressive decrease in their vision in 10 to 30 years [90]. However, in fundus images, fundus autofluorescence images, and fluorescein angiography, the exams appeared to be normal [89,90,91]. OMD is associated with the mutations in the *RP1L1* (*retinitis pigmentosa 1-like 1*) gene in an autosomal dominant fashion, while the same gene is also associated with autosomal recessive retinitis pigmentosa [91,92]. Given the normal findings in those exams, further exams, including OCT and electrophysiological testing, play a vital role in the diagnosis of OMD. In an OCT, OMD is associated with abnormal outer retinal changes, including the loss of the interdigitation zone in 100% of patients and a low reflectivity or discontinuous ellipsoid zone in the majority of patients [93].

The mfERG can better reflect the functional change across the OMD retina, which shows that the response densities significantly reduce in the central 7 degrees from the macula. The response densities gradually approach normal values toward the peripheral retina, while the implicit times of the mfERG are slightly delayed across the entire tested retinal area (the data were recorded by VERIS system, EDI Inc., San Mateo, CA, USA) [94]. In a focal ERG, OMD patients show a reduced response with a relatively smaller a-wave amplitude and a relatively larger b-wave amplitude [90]. In an ffERG and EOG, OMD patients have normal responses [95,96]. Therefore, mfERG and focal ERG are the two electrophysiological modalities that are vital in the workup of OMD.

### 2.7. North Carolina Macular Dystrophy

North Carolina macular dystrophy (NCMD, OMIM 136550) is a rare inherited macular dystrophy that is characterized by severe profound vision loss affecting patients as young as infants, but it is usually not progressive after around 12 years of age when the deterioration reaches a maximum [97]. Further vision loss, if any, tends to result from choroid neovascularization [98]. Clinically, NCMD can be divided into three grades by progressive macular changes: cluster of yellow-white lesions with little vision impairment (Grade 1), confluent of these yellow white lesions with moderate vision impairment (Grade 2), and central RPE and chorioretinal atrophy with severe central vision loss resembling congenital toxoplasmosis (Grade 3) [99]. It is an autosomal dominant disorder associated with mutations of the *PRDM13* gene, and causing the dysregulation of the retinal transcription factor PRDM13 [100,101]. NCMD Grade 1 patients can have deposits at the level of the RPE with intact ellipsoid zone. In severe NCMD patients, OCT can show abnormalities, including a central intrachoroidal fluid-containing space with preserved retinal layers [99]. Choroidal neovascularization and subretinal fibrosis can also be observed in NCMD patients with severely impaired vision [102].

In an ffERG, NCMD patients can appear to be normal [103]. However, delayed implicit times are seen in the entire central retina and reduced amplitudes are seen in the area of the lesion in the mfERG (the data were recorded by VERIS system, EDI Inc., San Mateo, CA, USA) [104]. In an EOG, a reduced LP:DT ratio from 1.4 to 1.6 in NCMD patients has been reported (the data were recorded by RETI-port/scan 21, Roland Consult, Wiesbaden, Germany) [103].

### 2.8. Pattern Dystrophy

Pattern dystrophy (PD) of the RPE is a heterogeneous group of autosomal dominant inherited retinal diseases. It has been proposed to include a group of disorders with distinct phenotypes, including AOFMD (discussed above), butterfly-shaped pigment dystrophy, Sjogren’s reticular type pattern dystrophy, pseudo-Stargardt pattern dystrophy, and fundus pulverulentus [105,106]. PD is characterized by a wide range of pigmented patterns in the macula, ranging from yellow-orange to grey-green [107]. These patterns are the result of the abnormal accumulation of lipofuscin in the RPE in various patterns and shapes [108]. These patterns can also be seen as hyperautofluorescence in fundus autofluorescence. PD most commonly affects bilaterally, and patients can present in a wide range of ages ranging from their 30s to 80s, with an average age of about 60 years old [107,109]. Most patients usually retain good vision, but severe vision loss can result from the formation of choroidal neovascular membranes or from geographic atrophy in the area of the lesion in all subtypes of PD [105,110,111]. PD is often associated with mutations in the *BEST1* (8% of patients) and *PRPH2* (27% of patients) genes in a large case series [109,111]. *PRPH2*, which is also known as retinal degeneration slow (RDS), encodes peripherin 2, a photoreceptor-specific transmembrane glycoprotein, which is critical for the formation of the outer segments of both rods and cones. Defective peripherin 2 can lead to disorganized outer segments, which eventually leads to the degeneration of photoreceptors [112]. Electrophysiological testing is a valuable tool in the workup of PD for ruling out other potential diagnoses. The specific patterns of RPE diseases with different results of ffERG and EOG are discussed below.

Butterfly-shaped pigment dystrophy (BPD, OMIM 169150) is a phenotype associated with a *PRPH2* mutation causing bilateral butterfly-shaped pigmented pattern in the macula area at the level of the RPE and abnormal EOG [113]. The patients may present without decreased visual acuity, with or without pigment mottling in the fundus exam in the early stage of the disease [108]. However, all patients, including those without decreased vision, have varying degrees of RPE atrophy. In photopic and scotopic ffERG as well as mfERG, the patients with BPD have been found to be normal [111,113]. The EOG LP:DT ratio is from a normal to reduced range from 1.3 to 1.6 [113,114].

Pseudo-Stargardt pattern dystrophy (PSPD, also known as multifocal pattern dystrophy) patients can present with decreased vision, worsening night vision, or metamorphopsia, which are similar to those with STGD1 [22]. However, patients with PSPD typically have only mild vision loss even in the advanced stage of disease, and have a better prognosis compared to STGD1 [24]. In the fundus exam, irregular yellow flecks can be found in the posterior pole similar to those in STGD1 [22]. However, in contrast to STGD1, PSPD patients do not have the “dark choroid” sign. In an ffERG, PSPD patients can have normal to severely reduced amplitudes in both photopic and scotopic conditions, which correspond to the confluence of the flecks [22]. The EOG LP:DT ratio can range from normal to severely reduced, with 55% of PSPD patients with abnormal EOG findings [22].

Sjogren’s reticular pattern dystrophy (RPD, OMIM 267800) is characterized by the presence of yellow deposits at the level of the RPE with a reticular pattern [115]. The pattern can be highlighted as a hyperfluorescent reticular net on FA [116]. The patient with RPD can present with normal or reduced vision. In electrophysiological testing, RPD patients have been reported to have normal scotopic and photopic ffERG [116,117,118]. RPD patients can have normal to mildly reduced EOG, with the LP:DT ratio ranging from 1.52–2.07 [116,117].

Fundus pulverulentus (FP) is the rarest form of PD, and only a few cases have been reported by now. The patients usually present in the fourth or fifth decade of life with mild and gradual loss of vision. First reported by Slezak and Hommer, it is characterized by a coarse granular appearance of the posterior pole with punctiform mottling in the RPE on the fundus exam [119,120,121]. Studies show that FP patients have normal ffERG and EOG results [119,121].

### 2.9. Central Areolar Choroidal Dystrophy

Central areolar choroidal dystrophy (CACD, OMIM 613105, 215500) is a rare inherited retinal disorder that affects the macula and is often associated with the atrophy of RPE and choriocapillaris within the central macula [122,123]. The patients typically present with bilaterally progressive loss of vision due to the dysfunction of the central photoreceptors with the age of onset between 27–58 years old [122,124,125]. In the fundus exam, CACD can be described in four stages [122,125]. Stage 1 is associated with minimal parafoveal pigmentary changes. Stage 2 is characterized by RPE alterations by round or oval atrophic hypopigmented areas from 1.5 to several disc diameters. Stage 3 is marked by the presence of one or more well-circumscribed areas of the RPE and choriocapillaris atrophy outside the fovea. When the area of the RPE and choriocapillaris atrophy involves the fovea, it is defined as Stage 4. CACD has been most commonly associated with mutations in the *PRPH2* gene with autosomal dominant inheritance [122,124,126]. The mutations in *GUCY2D* genes with autosomal dominant inheritance have also been linked to CACD [127]. Clinically, it is often difficult to distinguish CACD from atrophic age-related macular degeneration (AMD) or other inherited MDs, particularly without a clear family history. CACD patients are more likely to have a positive family history compared to AMD patients [128]. Therefore, electrophysiological testing can be valuable in the assessment of visual function and the diagnosis of CACD.

In the ffERG, CACD patients mostly have normal scotopic and photopic responses, even in advanced disease [122]. In some advanced CACD patients, a reduction in the ffERG amplitudes has been reported in photopic conditions but not scotopic conditions [123,125]. This indicates that the dysfunction in CACD is mainly on the cone pathway. In an mfERG, most CACD patients have reduced amplitudes and/or delayed implicit times, particularly in the parafoveal area. The reduction is also more pronounced in advanced stages (i.e., Stage 3 or 4) (the data were recorded by VERIS system, EDI Inc., San Mateo, CA, USA) [122,124]. In an EOG, most CACD patients have a normal (>2.0) LP:DT ratio, but a low LP:DT ratio has also been reported (the data were recorded by RETI-port/scan 21, Roland Consult, Wiesbaden, Germany) [122,126].

## 3. Conclusions

As the diagnosis of macular dystrophy becomes increasingly dependent on genetic testing and OCT findings, electrophysiological testing remains an essential tool for the accurate diagnosis and characterization of functional vision loss. As listed in Table 2 and Figure 4, different macular dystrophies generally present with abnormal a-wave and/or b-wave amplitudes and/or implicit times in the ffERG, given the effect of these processes on macular functions. However, in the diseases that affect more centrally, as in the case of BVMD, occult macular dystrophy, and North Carolina macular dystrophy, mfERG is necessary to detect the central reduction in mfERG amplitudes. Compared to ERGs, EOG tends to be less consistently affected in macular dystrophies, which can be used for differentiation from the diseases with defective RPE.

## Figures and Tables

**Figure 1 jcm-12-01430-f001:**
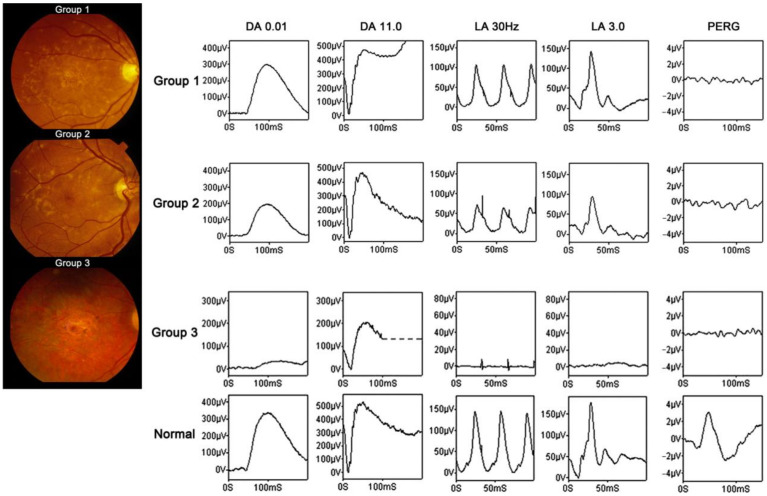
Representative fundus photos and electrophysiological findings in different groups of STGD1 (Adapted from Fujinami et al. 2013 with permission [26]). In Group 1, ffERG cone responses are mildly reduced and rod responses are normal, while PERG is reduced severely. In Group 2, ffERG cone responses are reduced moderately and rod responses are reduced mildly, while PERG is reduced severely. In Group 3, ffERG cone and rod responses are reduced severely, while PERG is reduced severely.

**Figure 2 jcm-12-01430-f002:**
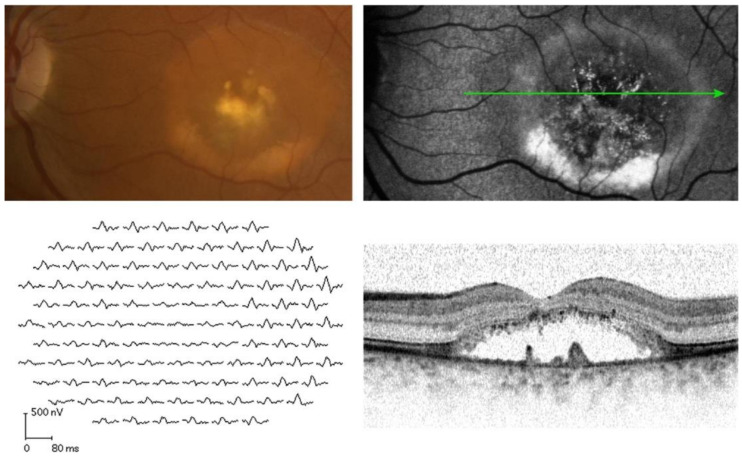
Morphologic and functional tests in a patient present with Best vitelliform dystrophy. (**Top left**): fundus image. (**Top right**): fundus autofluorescence with green arrow showing the direction of the OCT scan. (**Bottom left**): mfERG from the same patient. (**Bottom right**): OCT scan from the direction indicated by the green arrow as shown in (**top right**). The paracentral hyperautofluorescence in the image of fundus autofluorescence corresponds to the precipitate-like alterations on the outer surface of the photoreceptor outer segments. The hypoautofluorescence in the center of the lesion corresponds to atrophy and thinning of the outer nuclear layer in the OCT image. The focal paracentral accumulation on the retinal pigment epithelium/Bruch membrane shown in the fundus image and the image of fundus autofluorescence probably represents some degree of fibrosis. mfERG shows the significant reduction in amplitudes in the associated area (reused from Bitner et al. 2012 with permission [33]).

**Figure 3 jcm-12-01430-f003:**
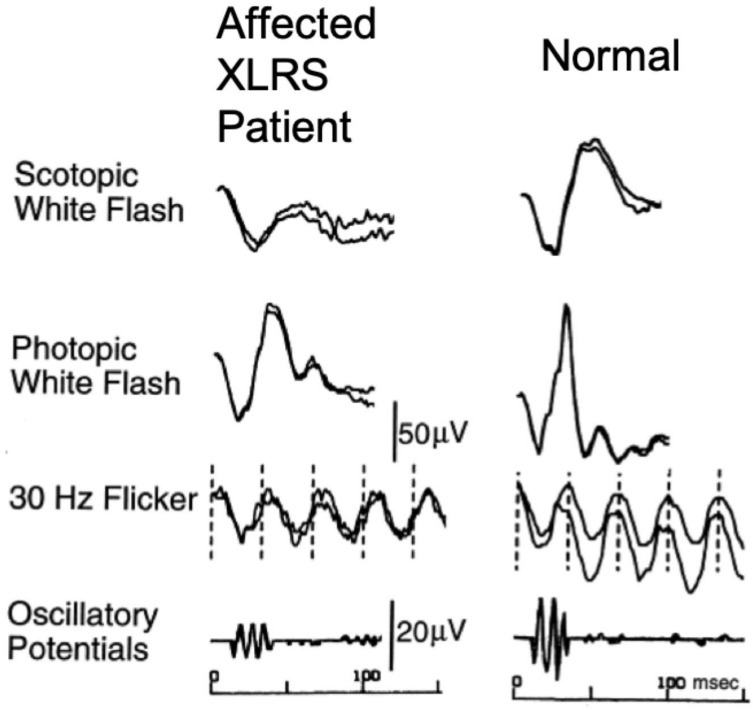
Electronegative response to white flash stimulation under scotopic condition in a XLRS patient (adapted from Sieving 1999 with permission [58]).

**Figure 4 jcm-12-01430-f004:**
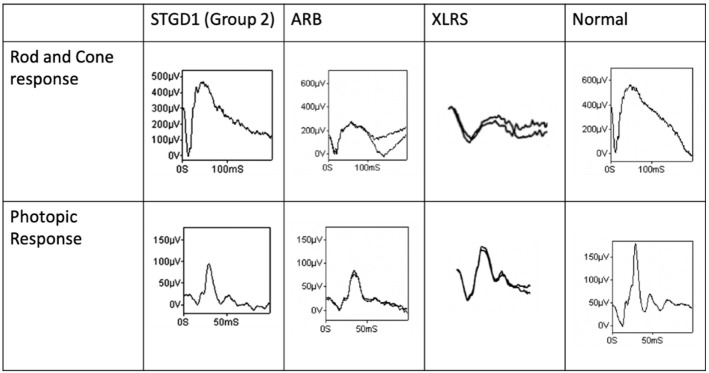
Abnormal ffERG waveforms of Stargardt disease (Group 2), autosomal recessive bestrinopathies, and X-linked retinoschisis compared to normal. The abnormalities of the ffERG waveforms caused by different extents of reduction in a- and b-waves under scotopic and photopic conditions are presented in these disorders (adapted from Fujinami et al. [26], Burgess et al. [50], and Sieving et al. [58] with permission).

**Table 1 jcm-12-01430-t001:** Genetic information of the macular dystrophies reviewed in this study.

NAME	OMIM	GENE	INHERITANCE
Stargardt disease 1	248200	*ABCA4*	Autosomal recessive
Stargardt disease 3	600110	*ELOVL4*	Autosomal dominant
Stargardt disease 4	603786	*PROM1*	Autosomal dominant
Best vitelliform macular dystrophy	153700	*BEST1*	Autosomal dominant
Adult-onset foveomacular vitelliform dystrophy	608161	*BEST1*, *IMPG1, IMPG2,* and *PRPH2*	Autosomal dominant
Autosomal dominant vitreoretinochoroidopathy	193220	*BEST1*	Autosomal dominant
Autosomal recessive bestrophinopathy	611809	*BEST1*	Autosomal recessive
X-linked juvenile retinoschisis	312700	*RS1*	X-linked
Sorsby fundus dystrophy	136900	*TIMP3*	Autosomal dominant
Doyne honeycomb retinal dystrophy	126600	*EFEMP1*	Autosomal dominant
Occult macular dystrophy	613587	*RP1L1*	Autosomal dominant
North Carolina macular dystrophy	136550	*PRDM13*	Autosomal dominant
Pattern dystrophies	169150	*PRPH2*	Autosomal dominant
Central areolar choroidal dystrophy	613105215500	*PRPH2,* and *GUCY2D*	Autosomal dominant

**Table 2 jcm-12-01430-t002:** Electrophysiology findings in macular dystrophies.

Disease	ffERG	mfERG	EOG	References
STGD1	A-	A-	A-	[25,26,28]
BVMD	nl	A-	A-	[33,37,39,40]
AOFMD	A-	A-	nl	[41,44,45]
ADVIRC	A-		nl-borderline	[46,47,48,49]
ARB	A-	A-	A-	[50,51,52]
XLRS	A-, I-	A-, I-	nl	[54,58,59,60,64,66]
SFD	nl-A-		A-	[72,76,77]
DHRD	A-	A-	nl	[82,85,86,87]
OMD	nl	A-	nl	[94,95,96,129]
NCMD	nl	A-, I-	A-	[103,104]
BPD	nl	nl	A-	[111,113,114]
PSPD	A-		A-	[22]
RPD	nl		A-	[116,117,118]
FP	nl		nl	[119,121]
CACD	nl-A-	A-, I-	nl	[122,124,125]

Abbreviations: A- Amplitude or LP:DT ratio; I- implicit time. Nl- normal.

## Data Availability

Not applicable.

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
