# Peer review of "Electrophysiological Evaluation of Macular Dystrophies"

_jcm, 2023, doi:10.3390/jcm12041430_

Round 1

Reviewer 1 Report

The manuscript needs to be thoroughly checked and revised for English grammar and typographical errors. Some of the errors are listed below.

Line 53_formatting error_patientsover

Line 86_formatting error_rethe duced

Line 223_incomplete sentence_On visual evoke...

Line 287_ The sentence seems to be incomplete.

The authors have described the electrophysiological characteristics of each type of macular dystrophy. It would benefit the audience if the authors can include a normal ERG vs characteristic ERG for each macular dystrophy, and then try to characterize each for clearly distinguishable type. (same as in table 2 but as figure presentation)

Author Response

The manuscript needs to be thoroughly checked and revised for English grammar and typographical errors. Some of the errors are listed below.

Line 53_formatting error_patientsover

Line 86_formatting error_rethe duced

Line 223_incomplete sentence_On visual evoke...

Line 287_ The sentence seems to be incomplete.re

 Re: Thank you for your valuable feedback. There are some formatting errors during edits for the manuscript and now all the formatting errors listed above have all been corrected. We also re-edited the entire manuscript to ensure formatting and grammatical mistakes are corrected.

The authors have described the electrophysiological characteristics of each type of macular dystrophy. It would benefit the audience if the authors can include a normal ERG vs characteristic ERG for each macular dystrophy, and then try to characterize each for clearly distinguishable type. (same as in table 2 but as figure presentation)

Re: Thank you for your valuable feedback. For STGD1, we have now included Figure 1 which shows ffERG data in normal and STGD1 in Group 1 to 3. We have also included mfERG etc of Best vitellifrom dystrophy in Figure 2. In Figure 3, we characterize the electronegative response in classic XLRS. Using Figure 4, we compare normal ffERG to abnormal ERGs in STGD1, ARB and XLRS as you suggested. For other more uncommon macular dystrophies such as SFD and DHRD, however, the ERG results were often described in original research articles with text without illustrations so we have difficulty getting more figures. In addition, the heterogeneity ERGs in different stages of disease make it difficult to illustrate the typical ERG waveforms in figure in some diseases, which can be better explained in text. 

Reviewer 2 Report

Chiang et al has comprehensively described the changes in electroretinographic responses in various macular dystrophies. This is very useful for the field for both clinicians and scientists to understand the abnormalities in the ERG waveforms and correlate with the disease. This will also help towards establishment of a uniform ERG protocol for screening macular dystrophies globally.

1. Macular dystrophies have been well reviewed in the manuscript individually, However basic introduction about ERG and use of ERG for clinical diagnosis of several retinal diseases is lacking. This review would benefit with a clear introduction of the method, representation of all the waveforms and the cell type specific response.

2. Also, correlation of abnormal ERG amplitude with the retinal cell type/region can even better elaborated for the readers to understand.

3. Line 223 is incomplete 

Author Response

Chiang et al has comprehensively described the changes in electroretinographic responses in various macular dystrophies. This is very useful for the field for both clinicians and scientists to understand the abnormalities in the ERG waveforms and correlate with the disease. This will also help towards establishment of a uniform ERG protocol for screening macular dystrophies globally.

  1. Macular dystrophies have been well reviewed in the manuscript individually, However basic introduction about ERG and use of ERG for clinical diagnosis of several retinal diseases is lacking. This review would benefit with a clear introduction of the method, representation of all the waveforms and the cell type specific response.

Re: Thank you for your fantastic feedback. We have added new content in Section 1 to briefly introduce and discuss the methods including ERG and other electrophysiologic testing methods discussed in the manuscript (e.g. EOG and PERG). We have also discussed the specific correlation between the components of electrophysiologic waveforms to different cell types as you suggested.  More details of the methods of electrophysiology can be added if needed.

  1. Also, correlation of abnormal ERG amplitude with the retinal cell type/region can even better elaborated for the readers to understand.

Re: Thanks for your suggestion. We agree that this would be helpful.  We have now further expanded on our discussion on the correlation between abnormal responses in ffERG to different cell types (e.g. rods and cones) and retinal layers (e.g. a-wave and b-wave) in all sections of the manuscript in both the method section (Section 1) and the section of application of ERG to different diseases (Section 2).

  1. Line 223 is incomplete 

Re: This has been fixed. Thank you for your comment.

Reviewer 3 Report

The authors of the “Electrophysiological evaluation of macular dystrophies “did collect and review recent publication regarding the functional electrophysiological findings in the macular dystrophies. 

General remarks

 Although the present material does have a potential to become an interesting and valuable contribution to clinical electrophysiology, I am not sure if in this format this review fulfills that role. A fundamental question for me after reading the manuscript is to whom this publication is intended. If it is designed to target a more research-oriented audience, the manuscript is very superficial in description and lacks depth. In that case I would expect a clear connection between specific molecular defect and its gene origin and how that is transfer to the electrophysiological readings. With chapters connecting specific defects, electrophysiological findings, and specific phenotypes of disease. But if assuming on the other hand that this is clinically oriented, then again, the manuscript does not fulfill that role. In that case I would expect a basic description of standard electrophysiological test, then lots of illustrations and examples presenting the most important clinically relevant electrophysiological findings with author’s opinion how good is the diagnostical value of this test. As in the current format the manuscript does not fulfill any of these roles.

Specific points

Independent from these general remarks I added a short list of potential problems in the current version of manuscript.

- The text would greatly benefit from spelling editing and careful deleting of double spaces.

- Not EVOL4   but ELOVL4

- missing: pattern dystrophy (Pseudostargardt), butterfly dystrophy, CACD

- pattern ERG is not the usual option in the clinical evaluation of photoreceptor diseases

- authors claim that b wave in retinal schisis is always smaller than a wave that is general direction but not always; sometimes the b wave is reduced so that it is similar as a wave

Author Response

The authors of the “Electrophysiological evaluation of macular dystrophies “did collect and review recent publication regarding the functional electrophysiological findings in the macular dystrophies.

General remarks

Although the present material does have a potential to become an interesting and valuable contribution to clinical electrophysiology, I am not sure if in this format this review fulfills that role. A fundamental question for me after reading the manuscript is to whom this publication is intended. If it is designed to target a more research-oriented audience, the manuscript is very superficial in description and lacks depth. In that case I would expect a clear connection between specific molecular defect and its gene origin and how that is transfer to the electrophysiological readings. With chapters connecting specific defects, electrophysiological findings, and specific phenotypes of disease. But if assuming on the other hand that this is clinically oriented, then again, the manuscript does not fulfill that role. In that case I would expect a basic description of standard electrophysiological test, then lots of illustrations and examples presenting the most important clinically relevant electrophysiological findings with author’s opinion how good is the diagnostical value of this test. As in the current format the manuscript does not fulfill any of these roles.

Re: We have now revised the manuscript to aim the article toward clinicians who are looking to apply electrophysiologic testing in their diagnosis of macular dystrophies. We have included figures from the most common MDs (Stargardt, BVD and XLRS) to demonstrate the electrophysiologic findings and the significance in these disease processes.

 Specific points

Independent from these general remarks I added a short list of potential problems in the current version of manuscript.

- The text would greatly benefit from spelling editing and careful deleting of double spaces. - Not EVOL4 but ELOVL4

Re: This has now been corrected.

- missing: pattern dystrophy (Pseudostargardt), butterfly dystrophy, CACD

Re: These have been modified and included in Section 2.8 and 2.9.

- pattern ERG is not the usual option in the clinical evaluation of photoreceptor diseases

Re: We have now removed PERG from Table 2 to reflect that, while PERG is still kept in the section about Stargardt disease due to part of the clinicians use it as additional evidence for diagnosis.

- authors claim that b wave in retinal schisis is always smaller than a wave that is general direction but not always; sometimes the b wave is reduced so that it is similar as a wave

Re: We have revised the content and added a sentence to address this point in the XLRS paragraph so that the content can cover a larger range of the ERG characteristics in retinoschisis.

Reviewer 4 Report

In this review, Chiang and Yu, summarizes the importance of electrophysiology testing in accurate diagnosis and characterization of functional vision loss in macular dystrophies, including Stargardt disease, Bestrophinopathies, X-linked retinoschisis, Sorsby fundus dystrophy, Doyne Honeycomb retina dystrophy, occult macular dystrophy, and North Carolina macular dystrophy. The manuscript is well written and tables are well presented. This review is a valuable addition to the field and has interesting observations which are beneficial to researchers in the areas of macular dystrophies and other inherited retinal disorders.

Major Corrections:

-          Line 34 – Add a paragraph on advantages of electrophysiology testing and explain the types of clinical ERG tests along with clinical relevance from various datasets generated (amplitudes, implicit times, OPs etc).

-          Throughout the manuscript, authors need to add the ERG company information in brackets for results mentioned in each of the macular dystrophies. This will be useful for readers and necessary for comparison of the data.

-          Line 223 – Uncompleted sentence

-          Line 224 – Throughout the manuscript, update “Donyne” to “Doyne”

-          Line 300 – Add few lines on the advances and future directions in diagnosis of macular dystrophies using ERGs.

Minor Corrections:

-          The authors need to add more specific references from original manuscripts throughout the manuscript.

-          The authors need to have a thorough read of the manuscript for spelling mistakes, spacing, and correct placement of abbreviations.

Author Response

In this review, Chiang and Yu, summarizes the importance of electrophysiology testing in accurate diagnosis and characterization of functional vision loss in macular dystrophies, including Stargardt disease, Bestrophinopathies, X-linked retinoschisis, Sorsby fundus dystrophy, Doyne Honeycomb retina dystrophy, occult macular dystrophy, and North Carolina macular dystrophy. The manuscript is well written and tables are well presented. This review is a valuable addition to the field and has interesting observations which are beneficial to researchers in the areas of macular dystrophies and other inherited retinal disorders.

Major Corrections:

- Line 34 – Add a paragraph on advantages of electrophysiology testing and explain the types of clinical ERG tests along with clinical relevance from various datasets generated (amplitudes, implicit times, OPs etc).

Re: We have added an entire section on review of different electrophysiologic testing and review of amplitudes, implicit times and OPs into the manuscript.

Minor Corrections:

- Throughout the manuscript, authors need to add the ERG company information in brackets for results mentioned in each of the macular dystrophies. This will be useful for readers and necessary for comparison of the data.

Re: We have added the equipment information of ffERG, mfERG, and EOG when they are available in the original research articles with abnormal data. However, there are a sizeable of references we cite in this review that do not include equipment information. We also did not include equipment information if the cited electrophysiologic results are normal (We can add them if you think it is necessary).

  • - Line 223 – Uncompleted sentence
  • - Line 224 – Throughout the manuscript, update “Donyne” to “Doyne”
  • - Line 300 – Add few lines on the advances and future directions in diagnosis of macular dystrophies using ERGs.

These have all been addressed and fixed. Thank you for your comments.

- The authors need to add more specific references from original manuscripts throughout the manuscript.

Re: We have added more original research papers as references to the manuscript.

- The authors need to have a thorough read of the manuscript for spelling mistakes, spacing, and correct placement of abbreviations.

Re: Thank you for your suggestions. We have read through the manuscript again and fixed mistakes as we can identify them.

Round 2

Reviewer 3 Report

The author responded to my question and reconstructed the manuscript to fit the clinical audience. In this respect I can recommend this manuscript for publishing.  

Reviewer 4 Report

In the revised manuscript, Chiang and Yu have thoroughly addressed all the initial concerns and the efforts are greatly appreciated. The revised review article can deserve publication without further changes.